# Insecticide resistance in *Aedes aegypti* from Tapachula, Mexico: Spatial variation and response to historical insecticide use

**Francisco Solis-Santoyo**[1], **Americo D. Rodriguez**[1†], **R. Patricia Penilla-Navarro**[1]*,
**Daniel Sanchez**[2], **Alfredo Castillo-Vera**[2], **Alma D. Lopez-Solis**[1], **Eduardo D. Vazquez-Lopez**[3], **Saul Lozano**[4], **William C. Black, IV**[5], **Karla Saavedra-Rodriguez**[5]

**1** Instituto Nacional de Salud Pública, Centro Regional de Investigación en Salud Pública, Tapachula, Chiapas, México, **2** El Colegio de la Frontera Sur, Unidad Tapachula, Tapachula, Chiapas, México, **3** Jurisdicción Sanitaria VII, Tapachula Chiapas, Antiguo Hospital General de Tapachula, Tapachula, Chiapas, México, **4** Centers for Diseases Prevention and Control, Arboviral Diseases Branch, Fort Collins, Colorado, United States of America, **5** Colorado State University, College of Veterinary Medicine and Biomedical Sciences Department of Microbiology, Immunology and Pathology, Arthropod Borne and Infectious Diseases Laboratory, Fort Collins, Colorado, United States of America

† Deceased.
* penilla@insp.mx

**Data Availability Statement:** All relevant data are within the manuscript and its Supporting Information files.

## Abstract

### Background

Insecticide use continues as the main strategy to control *Aedes aegypti*, the vector of dengue, Zika, chikungunya, and yellow fever. In the city of Tapachula, Mexico, mosquito control programs switched from pyrethroids to organophosphates for outdoor spatial spraying in 2013. Additionally, the spraying scheme switched from total coverage to focused control, prioritizing areas with higher entomological-virological risk. Five years after this strategy had been implemented, we evaluated the status and variability of insecticide resistance among *Ae. aegypti* collected at 26 sites in Tapachula.

### Methodology/Principal findings

We determined the lethal concentrations at 50% of the tested populations (LC$_{50}$) using a bottle bioassay, and then, we calculated the resistance ratio (RR) relative to the susceptible New Orleans strain. Permethrin and deltamethrin (pyrethroids), chlorpyrifos and malathion (organophosphates), and bendiocarb (carbamate) were tested. The frequencies of the substitutions V1016I and F1534C, which are in the voltage-gated sodium channel and confer knockdown-resistance (*kdr*) to pyrethroid insecticides, were calculated. Despite 5 years having passed since the removal of pyrethroids from the control programs, *Ae. aegypti* remained highly resistant to permethrin and deltamethrin (RR > 10-fold). In addition, following 5 years of chlorpyrifos use, mosquitoes at 15 of 26 sites showed moderate resistance to chlorpyrifos (5- to 10-fold), and the mosquitoes from one site were highly resistant. All sites had low resistance to malathion (< 5-fold). Resistance to bendiocarb was low at 19 sites, moderate at five, and high at two. Frequencies of the V1016I ranged from 0.16–0.71, while

**Funding:** ADR, PPN, WCBIV, and KSR received the award by the National Institute of Allergy and Infectious Diseases of the National Institutes of Health (https://www.niaid.nih.gov/) under Award Number R01AI121211 ("Insecticide Resistance Management to Preserve Pyrethroid Susceptibility in Aedes aegypti"). The content is solely the responsibility of the authors and does not necessarily represent the official views of the National Institutes of Health. The funders had no role in study design, data collection and analysis, decision to publish, or preparation of the manuscript.

**Competing interests:** The authors have declared that no competing interests exist. Author Americo D. Rodriguez was unable to confirm their authorship contributions. On their behalf, the corresponding author has reported their contributions to the best of their knowledge.

C1534 approached fixation at 23 sites (0.8–1). Resistance profiles and *kdr* allele frequencies varied across Tapachula. The variability was not associated with a spatial pattern at the scale of the sampling.

## Conclusion/Significance

Mosquito populations respond to selection pressure at a focal scale in the field. Spatial variation across sites highlights the importance of testing multiple sites within geographical regions.

## Author summary

*Aedes aegypti* is a major vector of dengue, chikungunya, Zika, and yellow fever. In the absence of effective vaccines or treatments, the suppression of mosquito populations using insecticides commonly has been implemented by public health programs. Unfortunately, few insecticides are available to target adult mosquitoes with outdoor spraying. The mismanagement of insecticides can lead to resistance selection in mosquitoes, affecting our ability to control arboviral diseases. Ideally, screening insecticide susceptibility in local mosquito populations allows public health entities to support insecticide management strategies that will prevent the development of resistance. In this study, we evaluated insecticide resistance in *Ae. aegypti* collected across 26 sites in Tapachula, Mexico. The results reveal the response of populations to its historic use of insecticides. High resistance to pyrethroids, used for 10 years, but not in the previous 5 years, and moderate to high resistance to chlorpyrifos, an insecticide from a different toxicological group and used for the past 5 years, were confirmed. High variation in resistance across *Ae. aegypti* sites suggests that focal selection plays an important role in the evolution of insecticide resistance in the field. Screening several collections sites within a geographical region provides better evidence to support strategies of insecticide management.

## Introduction

*Aedes aegypti* is the main vector of several arboviruses, including dengue, Zika, chikungunya, and yellow fever. The control of this mosquito species is challenging, mainly because it is highly adapted to urban and suburban areas and because it is widely dispersed in endemic regions [1]. Except for yellow fever, safe and effective treatments or vaccines for these diseases are still under study. Therefore, the suppression of *Ae. aegypti* remains the cornerstone to prevent transmission and control of outbreaks of these diseases [2].

Effective vector control involves several strategies, such as the elimination of potential breeding sites, application of chemical insecticides, and implementation of biological control. However, the application of chemical insecticides has become a common form of control because as a control is highly efficient and can be implemented promptly [3]. The most used insecticides in vector control have been the organophosphates temephos, used as a larvicide, and malathion, used as an adulticide by ultra-low volume application (ULV). Pyrethroids were introduced as adulticides in most Latin American countries in the 1990s [3]. In Mexico, according to the Mexican official policy for vector surveillance and control [4], the adulticide ULV formulation of permethrin, bioallethrin, and piperonyl butoxide (PBO) was used for more than 10 consecutive years (1999–2010). In the following 3–4 years, the pyrethroid d-

phenothrin + PBO was introduced. Subsequently, the use of organophospates returned in 2013, with chlorpyrifos and malathion being used as adulticides, while carbamates were recommended for indoor residual application [5].

The prolonged use of pyrethroid insecticides resulted in the evolution of resistance to them in *Ae. aegypti* worldwide, including Mexico, where failures in dengue control strategies are due in part to resistance [5]. Given that resistance to insecticides has been reported in populations of *Ae. aegypti* globally, the World Health Organization (WHO) recommends testing to ensure an effective insecticide management program. Decisions based on evidence of the resistance and/or susceptibility of *Ae. aegypti* will ensure a better selection of insecticides in vector control programs [6].

Two mechanisms of resistance to insecticides have been identified: resistance due to the enhanced metabolism of insecticides and insensitivity at the target site of the insecticides. Both mechanisms are involved in resistance to pyrethroids [7]. Knockdown resistance (*kdr*) refers to a phenomenon in which insects are not knocked down immediately after exposure to pyrethroids. *kdr* is caused by specific mutations at the voltage-gated sodium channel (VGSC), which is the target site for pyrethroids and DDT [8]. The amino acid substitutions V1016I [9], F1534C [10], and V410L [11,12] frequently have been associated with resistance to pyrethroids. Once these mutations are fixed in a population, reversion to susceptibility is difficult to achieve [9]. Therefore, the detection and characterization of *kdr* mutations in mosquito populations before resistance fixation occurs is essential for insecticide management strategies.

In Mexico, Chiapas is one of the states with the highest rate of endemic dengue cases. In particular, the city of Tapachula reports the highest incidence of dengue in the state [13], which is attributed to the proliferation of vectors that transmit emerging and re-emerging diseases. Under the region's tropical climate conditions, *Ae. aegypti* maintains high densities throughout the year. Consequently, dengue and other arboviruses transmitted by this vector have been prevalent in the region for a long time [14].

In the context of insecticide resistance management, we investigated the status of insecticide resistance to five insecticides, including two pyrethroids (permethrin and deltamethrin), two organophosphates (chlorpyrifos and malathion), and one carbamate (bendiocarb) and the spatial distribution of such resistance in populations of *Ae. aegypti* throughout the city of Tapachula. We expect that after 5 years of heavy use of organophosphates and the removal of pyrethroids from vector control campaigns, pyrethroid resistance will be lower, whereas organophosphate resistance will appear in focal points of the city. We tested 26 collection sites located in the city of Tapachula. Each collection site consisted of nine blocks and these were selected based on vehicle access for outdoor spraying. To minimize the effects of mosquito migration by flight range (50–150 m), sites were located at least 250 m apart. The spatial correlation between resistance and geographical distance was calculated for the 26 collection sites. In addition, since Tapachula's vector control program uses a quadrant subdivision for spraying activities, we included a second analysis to test this source of variation by assigning sites to one of the four cardinal geographical quadrants (NE, NW, SE, and SW).

## Materials and methods

### Collections

The study was conducted in the city of Tapachula, Chiapas, located in southern Mexico at 177 meters above sea level. The 26 collection sites located in four quadrants in the city: Northwest (NW), Northeast (NE), Southwest (SW), and Southeast (SE) are shown in Table 1. The biological material was collected from January to April 2018 using ovitraps of 1-L capacity [15]. Twelve ovitraps were installed at each collection site. Ovitraps were made by hand with

**Table 1. Geographic location of 26 *Aedes aegypti* collection sites in Tapachula, Chiapas, Mexico, in 2018.**

| Quadrant | Site | Neighborhood | Abbreviation | Latitude | Longitude |
|---|---|---|---|---|---|
| **Northeast** | | | | | |
| | NE-1 | Colinas del Rey | Col | 14˚55'50.9" | 92˚14'50.2" |
| | NE-2 | Galaxias | Gal | 14˚55'11.2" | 92˚15"06.5" |
| | NE-3 | Barrio Nuevo | Bar | 14˚54'51.0" | 92˚15'05.3" |
| | NE-4 | San Juan de los Lagos | SJL | 14˚54'26.3" | 92˚15'13.4" |
| | NE-5 | Coapantes | Coa | 14˚54'23.0" | 92˚14'57.1" |
| | NE-6 | Bonanza | Bon | 14˚54'02.8" | 92˚14'31.7" |
| | NE-7 | Centro (Country-Club) | CCC | 14˚54'22.7" | 92˚15'32.8" |
| **Southeast** | | | | | |
| | SE-1 | Galeana | Gal | 14˚54'00.2" | 92˚15'56.0" |
| | SE-2 | 16 de Septiembre | 16S | 14˚53'44.0" | 92˚15'42.1" |
| | SE-3 | Calcáneo Beltrán | Cal | 14˚53'28.0" | 92˚15'43.4" |
| | SE-4 | Benito Juárez 1 | BJ1 | 14˚53'21.8" | 92˚16'04.1" |
| | SE-5 | Benito Juárez 2 | BJ2 | 14˚53'11.7" | 92˚16'10.3" |
| | SE-6 | Emiliano Zapata | Zap | 14˚53'02.1" | 92˚16'14.2" |
| **Southwest** | | | | | |
| | SW-1 | Raymundo Enríquez | Ray | 14˚52'01.4" | 92˚18'48.8" |
| | SW-2 | Pobres Unidos | Pob | 14˚53'14.0" | 92˚17'6.1" |
| | SW-3 | Palmeiras | Pal | 14˚53'22.1" | 92˚18'06.4" |
| | SW-4 | Nuevo Milenio | Nue | 14˚53'24.8" | 92˚17'59.4" |
| | SW-5 | Primavera | Pri | 14˚53'39.3" | 92˚17'38.6" |
| | SW-6 | Democracia | Dem | 14˚54'23.7" | 92˚16'33.5" |
| **Northwest** | | | | | |
| | NW-1 | 5 de febrero | 5Fe | 14˚55'33.7" | 92˚15'22.4" |
| | NW-2 | Xochimilco 1 | Xo1 | 14˚55'48.9" | 92˚15'37.8" |
| | NW-3 | Xochimilco 2 | Xo2 | 14˚56'02.2" | 92˚15'29.9" |
| | NW-4 | Vergel 1 | Ve1 | 14˚56'21.2" | 92˚15'52.4" |
| | NW-5 | Vergel 2 | Ve2 | 14˚56'32.9" | 92˚15'52.4" |
| | NW-6 | Paraíso | Par | 14˚56'35.2" | 92˚15'19.7" |
| | NW-7 | Centro (Nva. España) | CNE | 14˚54'35.0" | 92˚15'43.5" |

transparent, inert, non-toxic polypropylene (PP) cups of a 1 L capacity, and painted black on the outside following the guidelines for Entomological Surveillance with Ovitraps of the Mexican Ministry of Health [15]. The interior of each ovitrap was lined with Whatman filter paper (No. 1) and filled with water to ¾ capacity; the paper was replaced weekly up to five times. The egg strips were transported to the insectary of the Regional Center for Research in Public Health/National Institute of Public Health (CRISP/INSP). The egg strips were submerged in 4 L of water in plastic containers (40 cm x 30 cm x 15 cm). On the third and sixth day, the hatched larvae were fed a diet of Harlan 5001 proteins, with 0.4 gr or 0.8 gr / 1.2 L for 1st-2nd stadium and 3rd-4th stadium at the 3rd and 6th day, respectively.

*Aedes aegypti* mosquitoes were identified to species and placed in cages (30 cm$^3$). Females were bloodfed from rabbit (under accepted procedures by the Ethical Commission of the Instituto Nacional de Salud Pública) to obtain the $F_1$ generation. Environmental conditions consisted of a temperature of 27 ± 2˚C, 70–80% humidity, and a 12:12 hour photoperiod. We used the insecticide-susceptible New Orleans reference strain of *Ae. aegypti*, provided by Dr. William Black and maintained in the CRISP/INSP insectary.

**Table 2. Concentrations (μg/bottle) used to determine the LC$_{50}$ of five different insecticides in the bottle bioassay for field *Aedes aegypti* and the susceptible reference strain.**

| Class | Mode of action | Insecticide | Concentration in μg/bottle | |
|---|---|---|---|---|
| | | | Field colonies | New Orleans reference |
| PYRs | sodium channel activators | Permethrin | 10, 20, 40, 80, 160 | 0.8, 1.2, 2.4, 3.2, 6 |
| | | Deltamethrin | 1, 2, 4, 6, 8, 16 | 0.75, 0.1, 0.15,0.2, 0.4 |
| OPs | cholinesterase inhibitors | Malathion | 2, 3, 4, 6, 8 | 2, 3, 4, 6, 8 |
| | | Chlorpyrifos | 2, 4, 6, 8, 12 | 0.2, 0.4, 0.8, 1.6, 3.2 |
| CARBs | | Bendiocarb | 0.5, 0.75, 1, 1.5, 3 | 0.25, 0.3, 0.4, 0.6, 1.2 |

PYRs = pyrethroids, OPs = organophosphates, CARBs = carbamates.

## Bioassays

The F$_1$ adults were exposed to the insecticides using a modified CDC bottle bioassay (Centers for Disease Control) [16]. Sigma brand technical grade insecticides were used to determine the lethal concentrations that killed 50% (LC$_{50}$) at each site. The pyrethroids permethrin (Type I) and deltamethrin (Type II), the organophosphates malathion and chlorpyrifos, and the carbamate bendiocarb were used to represent the toxicological groups used by vector control programs in Mexico.

To determine the LC$_{50}$, we tested five to six insecticide concentrations, which caused 10 to 90% mortality, in four replicates. Each insecticide LC$_{50}$ required approximately 500 mosquitoes. Table 2 shows the insecticide concentrations (μg/bottle) used to coat 250 ml Wheaton bottles using acetone as the solvent. During the bioassay, 15 to 20 (2–3 day old) females were gently aspirated into each bottle. The knockdown effect was recorded every 10 minutes for 1 hour. After 1 hour of exposure, the mosquitoes were transferred to plastic containers and maintained in the insectary to observe the mortality at 24 hours. The LC$_{50}$ of each insecticide was also determined for the susceptible New Orleans reference strain (NO) using a different set of insecticide concentrations (Table 2). Each insecticide LC$_{50}$ was replicated at least five times during a 7-month period. As control, a bottle impregnated only with acetone was used each time a test with field or susceptible mosquitoes was run.

The LC$_{50}$, 95% confidence intervals, slope, intercept, and *p* values were determined using the binary logistic regression model with QCal software [17]. The null hypothesis (Ho) assumes the observed mortality curve adjusts to a binary logistic regression model. Thus, we expected *p* values higher than 0.05 to accept the Ho. When the Ho was rejected, the bioassay was repeated.

To estimate the level of resistance among sites, a resistance ratio (RR) was calculated by dividing the LC$_{50}$ of the field sites by the LC$_{50}$ of the NO strain. The RR criterion according to Mazzarri and Georghiou [18] classifies high resistance as RR values greater than 10, moderate resistance as RR values between 5 and 10, and low resistance as RR values less than 5.

## Genotyping *kdr*-associated mutations

Genomic DNA was isolated from 50 F$_1$ individual female mosquitoes from each collection site following the method of Black and DuTeau [19]. The DNA was resuspended in TE buffer (10 mM Tris-HCl, 1 mM EDTA pH 8) and stored at -20˚C. The V1016I and F1534C mutations were genotyped according to the protocols of Saavedra-Rodríguez et al. [9] and Yanola et al. [10], respectively. The genotype and allelic frequencies were tested for Hardy-Weinberg (HW) equilibrium. The null hypothesis is that equilibrium is present in the population, which was verified with a chi-square test (df = 1 and *p* value > 0.05).

We tested the spatial variation of the $LC_{50}$s between the quadrants in the city using a linear model and ANOVA in R (car package). To test the hypothesis of resistance correlation with space, we created Moran's I correlograms as implemented in PASSaGE 2.0 [20]. Mosquitoes from different collection sites were considered neighbors if the sites were within 250 meters of each other. We expected that the $LC_{50}$s or *kdr* frequencies would be associated with geographical distance (i.e., that closer neighbor sites would show similar resistance levels, compared to those farther away). A second analysis to test the variation of the $LC_{50}$s and *kdr* frequencies between and within quadrants using a linear regression model and ANOVA in R (car package) was conducted. Since the city is uniformly sprayed during a cycle, we did not expect variation between or within quadrants. Correlation between *kdr* frequencies and $LC_{50}$s for permethrin and deltamethrin was tested using a Spearman test.

## Results

The geographic distributions of the resistance ratios (RR) for each insecticide in the 26 sites in Tapachula are shown in Fig 1. The $LC_{50}$ and confidence intervals for each of the five insecticides are shown in S1 Table. For the pyrethroids, we observed high levels of resistance widespread across sites. Fig 2A shows the permethrin RRs across Tapachula. High RRs were identified at 24 sites (RR from 11.4 to 43.1-fold). Only two sites—NE-3 and NW-2—showed moderate RRs (5.3 and 5.9-fold, respectively). The variation in RRs among quadrants was not significant (F = 0.56, df = 3, *p* value = 0.64). For deltamethrin, high RRs were determined in all 26 sites (10.6 to 101-fold). The variation among quadrants was not significant (F = 1.08, df = 3, *p* value = 0.37). Except for SW, all quadrants had at least one site with RR higher than 90-fold (Fig 2B).

The RRs for cholinesterase inhibitors (organophosphates and carbamates) are shown in Fig 3. For chlorpyrifos (Fig 3A), the RRs varied from low at 10 sites (0.68- to 4.9-fold) to moderate at 15 sites (5.2- to 7.2-fold) to high at one site (10.2-fold). No significant difference in RRs was found between quadrants (F = 1.08, df = 3, *p* value = 0.37). For malathion (Fig 3B), low resistance (0.86- to 4.5-fold) was identified at all 26 sites. However, a significant difference was observed between quadrants (F = 3.53, df = 3, *p* value = 0.03), with SE showing a mean RR of 2.6-fold (95% CI 1.9- to 3.2-fold). Resistance to bendiocarb was low (1.2- to 4.8-fold) at 19 sites, moderate (7.3- to 9.9-fold) at five sites, and high (10.3- to 11.2-fold) at two sites. No difference between quadrants was identified (F = 0.68. df = 3, *p* value = 0.57).

### *Kdr*-associated mutations

Genotype frequencies at the V1016I and F1534C loci in the voltage-gated sodium channel gene were determined in a sample of 45–50 individuals from each site (Table 3). The allele frequencies of the resistant allele I1016 fluctuated from 0.16–0.71. The lowest allele frequency (0.16) was scored for NE-3, whereas the highest frequency was from NW-6 (0.71). The remaining sites ranged from 0.2 to 0.5. Except for NE-2 and SW-4, the genotype frequencies at the V1016I loci were in HW equilibrium.

High allele frequencies of the resistant C1534 allele were determined at 22 of the 26 sites, ranging from 0.85 to 1.0. Lower frequencies (0.38–0.41) were found in NE-3, NW-5, and NW-7. While NE-7 was calculated with an intermediate value of 0.61. Most sites were in HW disequilibrium due to fixation of the resistant allele. We conducted a Spearman correlation test between the pyrethroid $LC_{50}$s and the expected frequencies of resistant homozygous genotypes. We found significant correlation coefficients among permethrin $LC_{50}$s, I1016/I1016 homozygotes (S = 2588, rho = 0.53, *p* value = 0.002), and C1534/C1534 homozygotes (S = 1966, rho = 0.515, *p* value = 0.004). Although it is known that C1534 shows protection

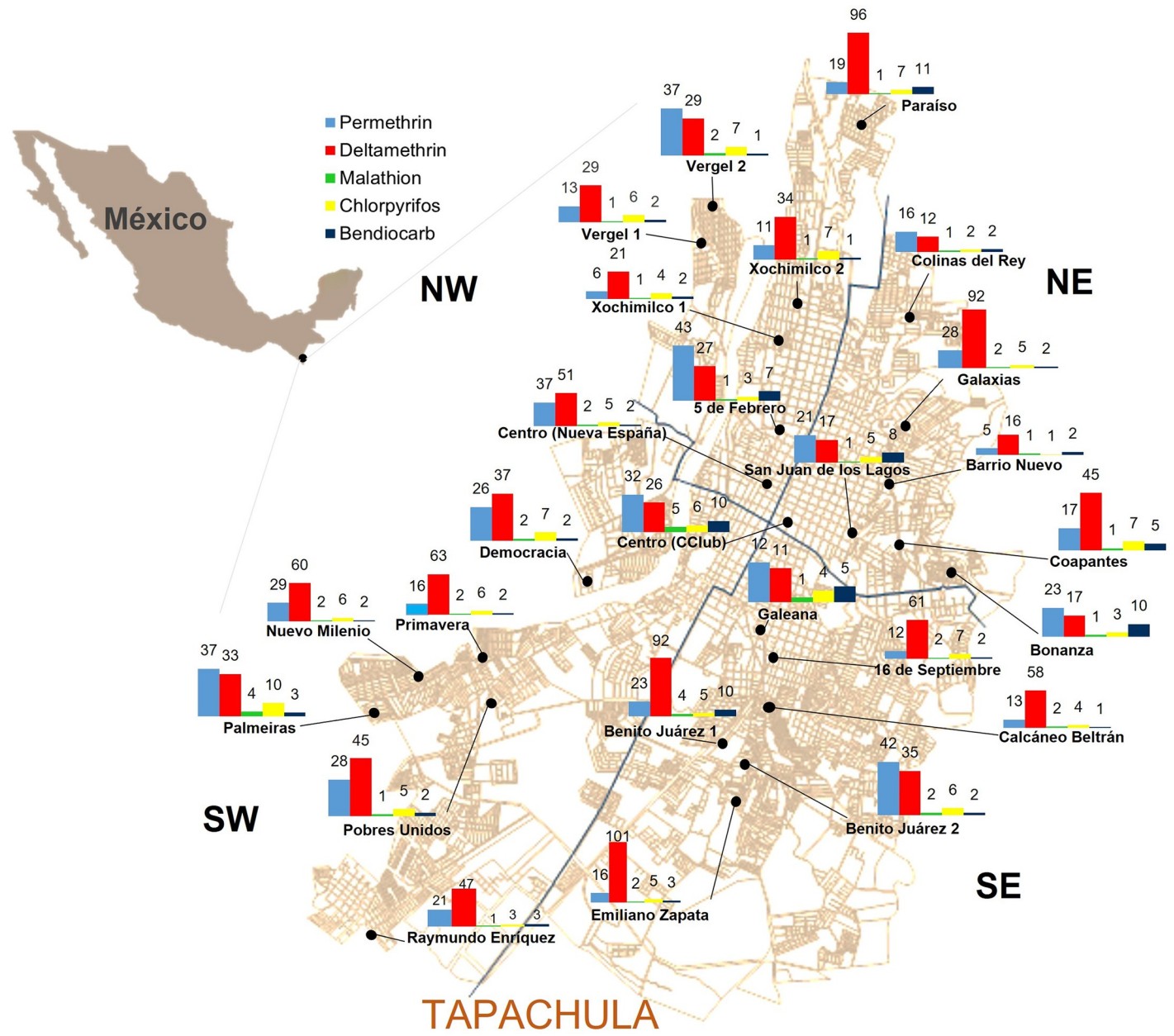

**Fig 1. Spatial distribution of insecticide resistance to five compounds in *Aedes aegypti* collected in Tapachula.** The number above each bar corresponds to the resistance ratio (RR). The RR was calculated relative to the susceptible New Orleans reference strain. Map obtained from the National Institute of Statistics and Geography (INEGI). Digital Map of Mexico. MDM: http://gaia.inegi.org.mx/mdm6.

only against permethrin (12), for deltamethrin a significant correlation was observed between the $LC_{50}$ and I1016/I1016 homozygotes (S = 2643, rho = 0.467, *p* value = 0.008) and the C1534/C1534 homozygotes (S = 2945, rho = 0.507, *p* value = 0.002). However, the significance for both insecticides disappeared when observations from the New Orleans reference strain were removed.

To assess the correlation of $LC_{50}$s with space, we generated Moran's I correlograms for each of the five insecticides (Fig 4). The analysis included all 26 collection sites. We did not detect a discernable pattern in any of the tested insecticides. We expected a positive correlation

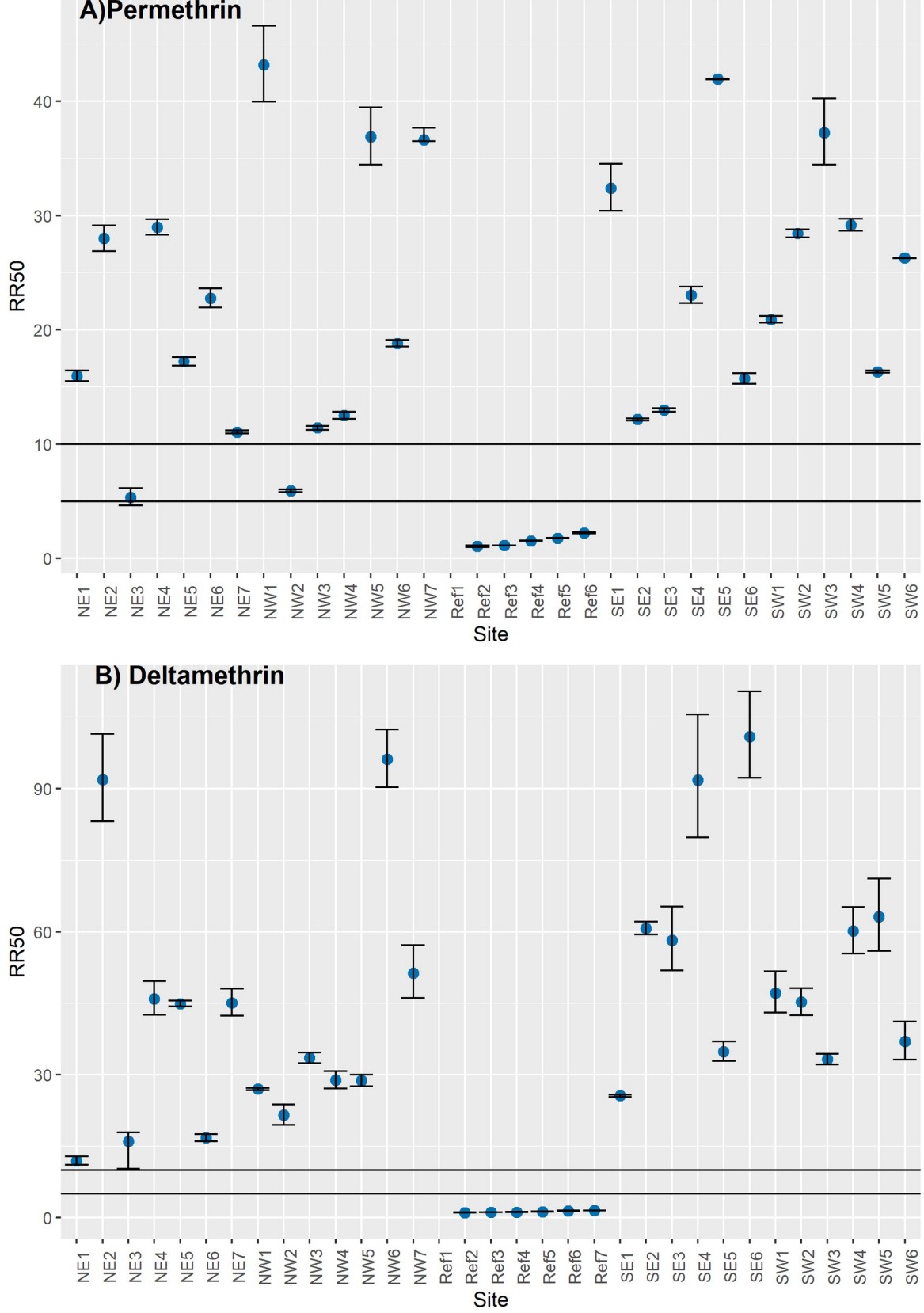

**Fig 2. Pyrethroids resistance ratios (RRs) of *Aedes aegypti* collected in 26 sites across Tapachula in 2018.** A) Permethrin and B) Deltamethrin. Dots represent the $RR_{50}$ with 95% confidence intervals for each site. Horizontal lines indicate the threshold for low resistance ($< 5$-fold), moderate resistance (5- to 10-fold) and high resistance ($> 10$-fold).

(Moran's I statistically $> 0$) between nearby sites, then as the distance increased (between the samples) the correlation would decrease, and later would turn negative (Moran's I statistically $< 0$). However, this was not observed. Although, few of the distance classes were statistically different from zero (eg. bendiocarb at 3250 m, 3750 m, 4750 m, and 6250 m; malathion at 1500 m; and deltamethrin at 3750 m, and 4250 m), a caveat in our analysis is the possibility that there is autocorrelation at smaller distances than the ones we selected (x $< 250$ m). Our experimental design was not geared towards the detection of spatial correlation at smaller distances; there were a small number of samples below 250 meters.

## Discussion

Efforts to control *Ae. aegypti* populations are hindered by widespread insecticide resistance worldwide. Local insecticide resistance monitoring is necessary for the design of specific and successful resistance management programs [21]. In Latin America, pyrethroids have been used for adult mosquito control since the 1990s. The switch to pyrethroids was based on environmental concerns that led to the use of less toxic classes of insecticides [22]. In Mexico, vector control programs implemented the use of permethrin in 1999 and continued their use until 2010. Local selection pressure caused a rapid evolution of pyrethroid resistance in *Ae. aegypti* across Mexico [9,23–27], resulting in policy modifications that recommended the use of insecticides with different toxicological modes of action.

In Tapachula, vector control programs replaced the use of permethrin with a different Type I pyrethroid (d-phenothrin + piperonyl butoxide) from 2010 to 2013. In 2013, pyrethroids were replaced by the organophosphate chlorpyrifos, and in 2017, by malathion. This study reveals the current status and response of local *Ae. aegypti* populations to these insecticide shifts. Despite the switch to organophosphates in the last 5 years, we observed that high levels of pyrethroid resistance remain widespread in Tapachula. An assumption in insecticide resistance management is that insecticide resistance has negative fitness costs. Therefore, when insecticide pressure is removed, populations are expected to reverse to susceptibility [28,29]. Currently, we are conducting a study to determine the degree of loss of resistance to pyrethroids from 2016 to 2020 in this study area, which will demonstrate whether mosquito populations in Tapachula are undergoing a process of decreasing resistance that will take several years. Another explanation is that pyrethroid resistance is maintained in *Ae. aegypti* populations by the domestic use of pyrethroids [30]. Surveys in Merida, Mexico, indicate that 85% of households took action to kill pests, and 89% exclusively targeted mosquitoes. Interestingly most of the aerosol spray cans contained pyrethroid insecticides [31].

Interestingly, RRs for deltamethrin—a Type II pyrethroid—were higher than permethrin RRs across sites. Deltamethrin was authorized by CENAPRECE for indoor residual use in 2009 for control of the malaria vector, but its use was restricted to rural areas. Therefore, direct selection pressure from the use of deltamethrin in public health is unlikely to be responsible for the high RRs in *Ae. aegypti* from Tapachula. Although all pyrethroids act at the same target site, the variability of resistance to their different types is attributed to different binding sites for Type I and Type II pyrethroids at the voltage-gated sodium channel. Additionally, the presence of enzymes that have a greater affinity to metabolize specific molecules within the same toxicological group might explain this variability [32].

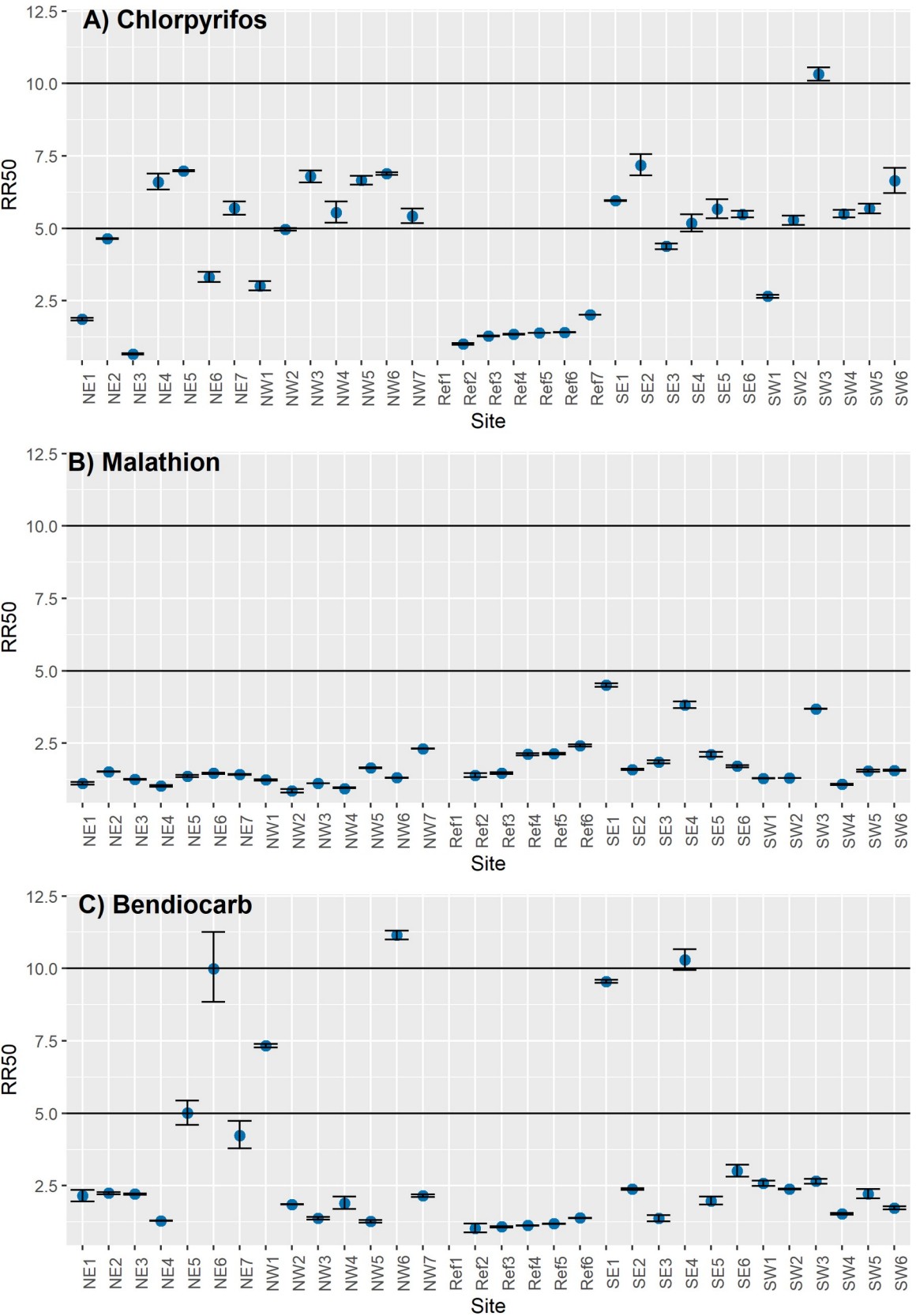

**Fig 3. Cholinesterase inhibitors resistance ratios (RRs) of *Aedes aegypti* collected in 26 sites across Tapachula in 2018.** A) Chlorpyrifos (organophosphate), B) Malathion (organophosphate), and C) Bendiocarb (carbamate). Dots represent the $RR_{50}$ with 95% confidence intervals for each site. Horizontal lines indicate the threshold for low resistance ($< 5$-fold), moderate resistance (5- to 10-fold) and high resistance ($> 10$-fold).

Knockdown resistance (*kdr*) is a major mechanism of pyrethroid resistance in *Ae. aegypti* from Mexico. In this study, we measured the frequency of this mechanism by molecular tests that identify mutations that confer changes to amino acids in the VGSC. The allele frequencies of the resistant allele I1016 ranged from 0.4 to 0.7, and for the resistant allele C1534, from 0.85 to 1.0 (except for three sites that had ~0.4). Historical data of *kdr* mutations indicated that C1534 confers low level of resistance on its own, and that resistance increased dramatically when I1016 evolved from the V1,016/C1,534 haplotype in field mosquito collected in different

**Table 3. Genotype counts and allele frequencies for two *kdr*-associated substitutions (V1016I and F1534C) from *Aedes aegypti* collected at 26 sites in Tapachula.** RR = homozygote resistant, RS = heterozygote, and SS = homozygote susceptible. * indicates a lack of Hardy-Weinberg equilibrium.

| Site | Abv | N | V1016I genotypes | | | I1016 frequency | F1534C genotypes | | | C1534 frequency |
|------|-----|---|------|------|------|-----------------|------|------|------|-----------------|
| | | | I/I | V/I | V/V | | C/C | F/C | F/F | |
| | | | RR | RS | SS | | RR | RS | SS | |
| NE-1 | Col | 48 | 4 | 23 | 21 | 0.32 | 47 | 1 | 0 | 0.99 |
| NE-2 | Gal | 48 | 5 | 30 | 13 | 0.42* | 39 | 8 | 1 | 0.90 |
| NE-3 | Bar | 48 | 1 | 13 | 34 | 0.16 | 7 | 22 | 19 | 0.38 |
| NE-4 | SJL | 48 | 6 | 22 | 20 | 0.35 | 47 | 1 | 0 | 0.99 |
| NE-5 | Coa | 45 | 5 | 24 | 16 | 0.38 | 42 | 3 | 0 | 0.97 |
| NE-6 | Bon | 48 | 8 | 22 | 18 | 0.4 | 43 | 5 | 0 | 0.95 |
| NE-7 | CCC | 50 | 15 | 25 | 10 | 0.55 | 11 | 39 | 0 | 0.61* |
| | Subtotal | 335 | 44 | 159 | 132 | 0.37 | 236 | 79 | 20 | 0.82* |
| SE-1 | Gal | 48 | 12 | 28 | 8 | 0.54 | 42 | 6 | 0 | 0.94 |
| SE-2 | 16S | 48 | 5 | 29 | 14 | 0.4 | 48 | 0 | 0 | 1* |
| SE-3 | Cal | 48 | 7 | 25 | 16 | 0.41 | 46 | 2 | 0 | 0.98 |
| SE-4 | BJ1 | 48 | 4 | 28 | 16 | 0.38 | 46 | 2 | 0 | 0.98 |
| SE-5 | BJ2 | 48 | 11 | 25 | 12 | 0.49 | 47 | 1 | 0 | 0.99 |
| SE-6 | Zap | 48 | 10 | 25 | 13 | 0.47 | 34 | 14 | 0 | 0.85 |
| | Subtotal | 288 | 49 | 160 | 79 | 0.45 | 263 | 25 | 0 | 0.96 |
| SW-1 | Ray | 48 | 5 | 28 | 15 | 0.39 | 48 | 0 | 0 | 1* |
| SW-2 | Pob | 48 | 14 | 21 | 13 | 0.51 | 47 | 1 | 0 | 0.99 |
| SW-3 | Pal | 48 | 2 | 17 | 29 | 0.22 | 46 | 0 | 2 | 0.96* |
| SW-4 | Nue | 48 | 0 | 26 | 22 | 0.27* | 48 | 0 | 0 | 1* |
| SW-5 | Pri | 48 | 8 | 22 | 18 | 0.39 | 47 | 1 | 0 | 0.99 |
| SW-6 | Dem | 48 | 13 | 20 | 15 | 0.48 | 41 | 7 | 0 | 0.93 |
| | Subtotal | 288 | 42 | 134 | 112 | 0.38 | 277 | 9 | 2 | 0.98* |
| NW-1 | 5Fe | 48 | 9 | 22 | 17 | 0.42 | 48 | 0 | 0 | 1* |
| NW-2 | Xo1 | 48 | 10 | 29 | 9 | 0.51 | 48 | 0 | 0 | 1* |
| NW-3 | Xo2 | 48 | 1 | 21 | 26 | 0.24 | 48 | 0 | 0 | 1* |
| NW-4 | Ve1 | 48 | 11 | 23 | 14 | 0.47 | 48 | 0 | 0 | 1* |
| NW-5 | Ve2 | 50 | 8 | 17 | 25 | 0.33 | 2 | 37 | 11 | 0.41* |
| NW-6 | Par | 48 | 24 | 20 | 4 | 0.71 | 48 | 0 | 0 | 1* |
| NW-7 | CNE | 50 | 4 | 26 | 20 | 0.34 | 0 | 38 | 12 | 0.38* |
| | Subtotal | 340 | 67 | 158 | 115 | 0.43 | 242 | 75 | 23 | 0.82* |
| Total | | 1251 | 202 | 611 | 438 | 0.41 | 1018 | 188 | 45 | 0.89* |

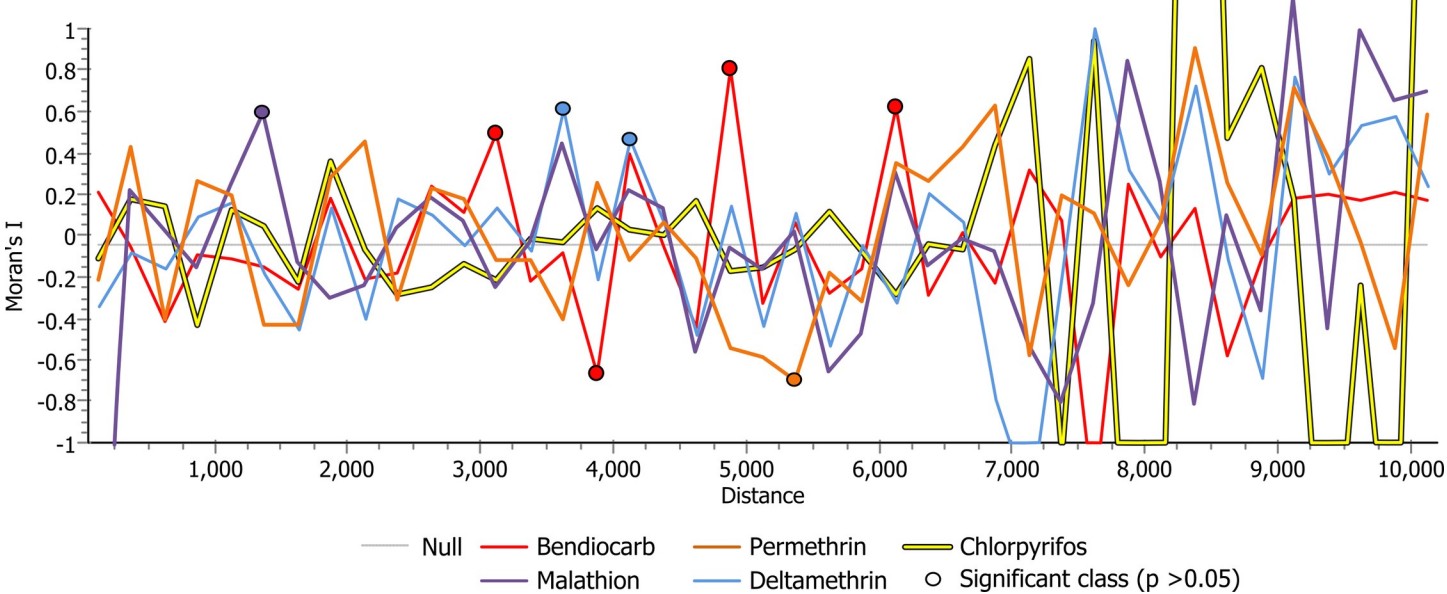

**Fig 4. Moran's I correlograms as implemented in PASSaGE 2.0 assessing the correlation of LC$_{50}$s with space for permethrin (pyrethroid), deltamethrin (pyrethroid), chlorpyrifos (organophosphate), malathion (organophosphate), and bendiocarb (carbamate).** The analysis included 26 collection sites in Tapachula, Chiapas, Mexico. *Aedes aegypti* mosquitoes from different collection sites were considered neighbors if the sites were within 250 meters of each other.

places from Mexico [33]. Those results demonstrated that I1016 was unlikely to have evolved independently, and that both mutations need coexist in the same mosquito in order to confer higher levels of resistance. Moderate correlations were significant between the resistant allele frequencies and the RRs for permethrin and deltamethrin only when including the New Orleans datapoints. This significance might be explained by most of the allele frequencies being distributed within a small range of variability.

This study was conducted after chlorpyrifos had been used for 5 years in outdoor spraying by vector control programs. Our results provide evidence of the response of *Ae. aegypti* populations to chlorpyrifos pressure. Ten sites showed low RRs, 15 sites showed moderate resistance, and one site was highly resistant. Interestingly, *Ae. aegypti* from all 26 sites were susceptible or had low RRs to malathion, thereby indicating that resistance to chlorpyrifos does not predict the lack of effectivity of malathion. Additionally, the RRs to bendiocarb were variable: mosquitoes from 19 sites had low RRs, those from three were moderate, and those from two were highly resistant. Only a few sites showed moderate to high resistance to both chlorpyrifos and bendiocarb (NE-5, NW-6, and SE-4). The lack of cross-resistance between organophosphates and carbamates suggests that the resistance mechanisms are not due to the insensitivity of their target site (the acetylcholinesterase) [34] and, in fact, no mutations have been found in *ace-1* gene in *Aedes aegypti* [35].

A survey in Veracruz, Mexico, identified high RRs to chlorpyrifos in Cosoleacaque (RR = 13.9), moderate RRs in Poza Rica (RR = 7.9), and low RRs in five sites in Veracruz [36]. By using a discriminating dose of 50 μg/bottle and 85 μg/bottle for 30 minutes, two additional studies were able to identify chlorpyrifos resistance in Mexico [26,37]. Since neither of these studies found a history of chlorpyrifos use in vector control programs, the resistance might be explained by indirect exposure to chlorpyrifos through the extensive use of this insecticide to control agricultural pests [36].

During vector control programs, the city of Tapachula is uniformly sprayed, using the same insecticide, frequency and intensity. More yet, we selected sites based in their accessibility to

spraying-vehicles. Assuming that no spatial heterogeneity in frequency and intensity of spraying, we did not expect the high levels of variation in resistance profiles across the city. For example, significant heterogeneity in the frequency of *kdr* haplotypes was detected in *Ae. aegypti* collected between city blocks in a town of Yucatan, suggesting that selection for these haplotypes occurs at a fine spatial scale (37). However, in contrast to our study, insecticide application was highly variable in space and time, creating a mosaic of selection pressures. In our study, some sources of heterogeneity could occur from mosquito migration from untreated sites due to vehicle inaccessibility, including parks, cemeteries, steep and unpaved streets. A second source of insecticide pressure is by use of household aerosol insecticides. For example, in a previous study from Merida, Mexico approximately 87% of households used commercially available pyrethroid products to control mosquitoes in their homes (31). Future studies should include an assessment of this source of selection pressure in Tapachula.

The spatial variability in insecticide resistance observed across the 26 sites in Tapachula is likely associated with the presence or appearance of "hot spots or dengue foci," which contribute to the persistent transmission of the diseases and therefore to focal areas with greater spray intervention [38]. In addition, the spatial variability of resistance highlights the importance of evaluating resistance in multiple sites within a defined geographic area for the application of appropriate vector control decisions. Although no geographical correlation/association/pattern between resistance was found in Tapachula, more specific and finer environmental characteristics must not be discarded in future studies. A previous study used mitochondrial ND4 haplotypes to determine gene flow patterns among 38 *Ae. aegypti* coastal collections in Mexico [39]. Three genetic clusters were identified, the Northeast, Pacific, and Yucatan peninsula. For all sites, genetic distances remained small below geographic distances of 90 km and became large at distances >150 km. The Pacific cluster had the highest gene flow and diversity. A second study in the Yucatan Peninsula showed high gene flow occurring across 27 *Ae. aegypti* collection sites located up to 150 km of distance. Single nucleotide polymorphism (SNPs) at eleven loci did not vary across sites, suggesting high levels of gene flow. In contrast, insecticide resistance loci, including *kdr* alleles (I1016 and C1534) were highly variable across sites, indicating that insecticide resistance offsets the homogenizing effects of gene flow [40]. In this study, we assume complete gene flow among collection sites because: 1) Tapachula belongs to the Pacific cluster, 2) *Ae. aegypti* is well established throughout the year and, 3) collection sites are within 10 km of distance. However, this remains to be tested.

## Conclusion

Despite more than 5 years having passed since the removal of pyrethroids from vector control programs in Tapachula, high levels of pyrethroid resistance and *kdr*-associated alleles persist in *Ae. aegypti* populations. Future resistance surveys will reveal if pyrethroid resistance is maintained in mosquito populations. We observed that, after 5 years of chlorpyrifos use in vector control programs, more than 50% of the sites have moderate to high chlorpyrifos resistance but complete susceptibility to malathion. Since malathion was introduced later in 2017, future studies to evaluate the selection of malathion resistance in the field are needed. Two different analyses were conducted 1) the spatial analysis included all 26 sites and, 2) the quadrant analysis to identify operational sources of heterogeneity. The quadrant analysis doesn't include a geographical component and has limitations. Insecticide resistance varied spatially, most likely as a consequence of the pattern of insecticide use combined with environmental factors. Based on the results of our study, we suggest that both of the studied organophosphates and the carbamate remain viable options for use in the control strategy for this vector. The return to a pyrethroid (at least permethrin and deltamethrin) for outdoor spraying is recommended when

the levels of resistance have decreased to RR less than 10-fold and once mechanisms other than *kdr* have been elucidated for pyrethroid resistance.

## Supporting information

**S1 Table. Lethal concentration to kill 50% (LC$_{50}$) for five insecticides at 26 *Aedes aegypti* sites in Tapachula, Chiapas, Mexico.** The LC$_{50}$ is in micrograms (μg) of active ingredient per bottle. The 95% confidence intervals around the LC$_{50}$ are enclosed in parentheses. *p* values higher than 0.05 indicate the observed data fit the binary logistic regression model. (XLSX)

**S2 Table. Genotype for two *kdr*-associated substitutions (V1016I and F1534C) per individual *Aedes aegypti* mosquito (n = 47 to 50) at 26 sites in Tapachula, Chiapas, Mexico.** For V1016I: AA = homozygote for Ile$_{1016}$, resistant; GG = homozygote for Val$_{1016}$, susceptible; AG = heterozygote Ile$_{1016}$/Val$_{1016}$. For F1534C: TT homozygote for Phe$_{1534}$, susceptible; GG = homozygote for Cys$_{1534}$, resistant; TG = heterozygote Phe$_{1534}$/Cys$_{1534}$. (XLSX)

## Acknowledgments

The authors would like to thank Biol. Elsa Contreras, Geovanni Vázquez and Francisco Pozos from the Programme of Vector Control in Tapachula for contributions in the field work.

## Author Contributions

**Conceptualization:** Americo D. Rodriguez, R. Patricia Penilla-Navarro, William C. Black, IV.

**Data curation:** Francisco Solis-Santoyo, Alfredo Castillo-Vera, Alma D. Lopez-Solis, Karla Saavedra-Rodriguez.

**Formal analysis:** Francisco Solis-Santoyo, Americo D. Rodriguez, Alma D. Lopez-Solis, Saul Lozano, William C. Black, IV, Karla Saavedra-Rodriguez.

**Funding acquisition:** Americo D. Rodriguez, William C. Black, IV.

**Investigation:** Francisco Solis-Santoyo, Americo D. Rodriguez, R. Patricia Penilla-Navarro, Daniel Sanchez, Alma D. Lopez-Solis, Saul Lozano, William C. Black, IV, Karla Saavedra-Rodriguez.

**Methodology:** Francisco Solis-Santoyo, Americo D. Rodriguez, Daniel Sanchez, Alma D. Lopez-Solis, Eduardo D. Vazquez-Lopez, Saul Lozano, Karla Saavedra-Rodriguez.

**Project administration:** Americo D. Rodriguez, Alma D. Lopez-Solis, William C. Black, IV, Karla Saavedra-Rodriguez.

**Software:** Saul Lozano.

**Supervision:** Americo D. Rodriguez, R. Patricia Penilla-Navarro, Daniel Sanchez, Alfredo Castillo-Vera, Eduardo D. Vazquez-Lopez, Karla Saavedra-Rodriguez.

**Validation:** Americo D. Rodriguez, R. Patricia Penilla-Navarro, Alfredo Castillo-Vera, Alma D. Lopez-Solis, Saul Lozano, Karla Saavedra-Rodriguez.

**Visualization:** Francisco Solis-Santoyo, Eduardo D. Vazquez-Lopez.

**Writing – original draft:** Francisco Solis-Santoyo.

**Writing – review & editing:** Americo D. Rodriguez, R. Patricia Penilla-Navarro, Daniel San-
   chez, Alfredo Castillo-Vera, Alma D. Lopez-Solis, Eduardo D. Vazquez-Lopez, Saul Lozano,
   William C. Black, IV, Karla Saavedra-Rodriguez.

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
