## [Decision Letter · Decision Letter 0]

8 Apr 2021

Dear Dr. Penilla-Navarro,

Thank you very much for submitting your manuscript "Insecticide resistance in Aedes aegypti from Tapachula, Mexico: spatial variation and response to historical insecticide use" for consideration at PLOS Neglected Tropical Diseases. As with all papers reviewed by the journal, your manuscript was reviewed by members of the editorial board and by several independent reviewers. The reviewers appreciated the attention to an important topic. Based on the reviews, we are likely to accept this manuscript for publication, providing that you modify the manuscript according to the review recommendations. 

Sincerely,

Pattamaporn Kittayapong, Ph.D.

Associate Editor

Amy Morrison, Ph.D.

Deputy Editor

Reviewer's Responses to Questions

**Key Review Criteria Required for Acceptance?**

**Methods**

-Are the objectives of the study clearly articulated with a clear testable hypothesis stated?

-Is the study design appropriate to address the stated objectives?

-Is the population clearly described and appropriate for the hypothesis being tested?

-Is the sample size sufficient to ensure adequate power to address the hypothesis being tested?

-Were correct statistical analysis used to support conclusions?

-Are there concerns about ethical or regulatory requirements being met?

Reviewer #1: Study Objectives: Yes, clearly stated – to test for resistance to five different insecticides of three different chemical classes, and to perform spatial mapping of resistant mosquitos. 

Hypothesis: Yes, clearly stated – pyrethroid resistance is predicted to be lower, while organophosphate resistance is predicted to increase at specific focal points in Tapachula. 

Study Population: Yes, clearly described and appropriate. The study is of Aedes aegypti populations in Tapachula, the city with the highest incidence of dengue in Chiapas state, making the results of this study of potentially high public health impact. 

Study Design

Critiques:

- Line 153: “Females were bloodfed to obtain the F1 generation.” Please specify the animal species on which the females were bloodfed.

- Lines 198-199: The authors describe the use of city quadrants as the basis for some of their spatial variation analysis, however this seems like a somewhat arbitrary geographical division on which to perform spatial analysis. For instance, two collection sites may be very close in linear distance, but are located in different quadrants, leading to underestimation of spatial effects between quadrants. While it is okay to keep the quadrant spatial analysis in the paper, I recommend they discuss the limitations of this approach in more detail in the Discussion section.

- Lines 200-207: The authors describe a correlation analysis (using Moran’s correlograms) testing the hypothesis that mosquitos from neighboring collection sites will have more similar LC50s than mosquitos from distant sites, however I do not see the results of this analysis in the Results or Discussion section. Please include this data in the revised manuscript (even if the results are negative or placed in Supplemental Data), as they provide more robust and meaningful spatial information than the quadrant-based approach. 

Strengths:

- A major strength of the study is the integration of spatial, phenotypic (insecticide resistance, as measured by bottle bioassay), and genotypic data. 

- The geographic locations of the Aedes collection sites are precisely and thoroughly described.

Statistics: Yes, statistical approach is sound overall with sufficient sample sizes. 

Ethical and Regulatory: No concerns.

Reviewer #2: Methods are written clearly, sample sizes are adequate, analyses are appropriate overall.

Reviewer #3: Yes, the objectives were clearly articulated with clear hypothesis stated. The study design, population description and choice, sample size, and statistical analysis were appropriate for this type of study. I have no reservations about ethical or regulatory issues. 

I had three specific critiques of the methods:

-Line140: What was the material and color of the ovitraps used? Where they a specific brand or homemade?

-Line 146: At what approximate concentration (grams of diet per pan, etc.) and at what frequency were the larvae fed?

-Lines 165-175: What was the control treatment used for the LC50 testing? Was acetone alone used? Were the control mosquitoes untreated? Both? Please clarify.

**Results**

-Does the analysis presented match the analysis plan?

-Are the results clearly and completely presented?

-Are the figures (Tables, Images) of sufficient quality for clarity?

Reviewer #1: The analysis presented does not match the stated analysis plan, as I do not see any results for correlating LC50 profile similarity with distance to other collection sites (Lines 200 to 207), as discussed above. 

The results are otherwise clearly and completely represented. The figures are of sufficient visual quality and clarity.

Reviewer #2: Line 260. Table 3. It would be useful to describe the mutation status per individual mosquito (raw data as supplement) and a summary in the results for the two mutation sites. It is not likely that the mutations are evolving or acting independently.

Line 269 - What is known about these particular mutations and insensitivity towards Type I vs Type II pyrethroids? 

Line 275 - The significance is said to have disappeared, but the sentence is ambiguous and the reader cannot tell whether this is just for the deltamethrin analysis or the permethrin analysis as well. This is clarified in the Discussion, but it would be better to clarify at this first instance where it is mentioned.

Line 337 - Target-site mutations (acetylcholinesterase) are known to be unlikely in Aedes aegypti, so this is not an unexpected result. A reference about this should be included here.

Line 352 - No geographical correlation with resistance patterns were found. Is anything known about the genetic population structure or history of colonisation by Ae. aegypti in Mexico? Resistance alleles can be brought in with mosquitoes invading from other places. Is this a likely scenario in Tapachula? In some cases it might explain the presence of resistance where there is no local history of use of a particular insecticide in the local area.

Reviewer #3: The analyses, results, tables, and figures were appropriate and clearly presented the data. 

I have a specific critique of Figure 1:

-Please clarify the location of ALL the sites by placing a dot indicating each site. Some sites (e.g. Raymundo Enriquez, Paraiso, etc.) have the resistance ratios placed adjacent to the site, but the site location is not clear.

**Conclusions**

-Are the conclusions supported by the data presented?

-Are the limitations of analysis clearly described?

-Do the authors discuss how these data can be helpful to advance our understanding of the topic under study?

-Is public health relevance addressed?

Reviewer #1: Please address the limitations of the quadrant-based spatial analysis, as detailed above.

Yes, the conclusions are supported by the presented data. The authors do a good job discussing how the data advance the field and explicitly state how their findings impact Tapachula and mosquito control more generally. 

Minor points to clarify:

- Discuss the possible contribution of regional/spatial heterogeneity in the intensity/frequency or type of insecticide used in Tapachula that can explain the spatial variation in resistance profiles. It sounds like insecticides were used uniformly throughout the city, adhering to national guidelines, but it would be helpful for the authors to include these potential contributing factors in the Discussion. 

- Discuss how much population mixing you would expect between Aedes in neighboring collection sites. How frequently and how far does Aedes tend to migrate between different areas in urban habitats? How does that inform how you interpret the spatial variation you observe in your phenotypic and genotypic data?

Reviewer #2: It would be useful to discuss the VGSC results in light of the hypotheses of co-evolution of Ile1016 and Cys1534 in Mexico put forward by Vera-Maloof et al. 2015 (PlosNTD)

Reviewer #3: The conclusions appear to be correctly founded on the literature and supported by the data presented. The limitations and recommendations for future action are clearly described and appropriate. The relevance to the advancement of the field and to public health are addressed.

**Editorial and Data Presentation Modifications?**

Reviewer #1: None.

Reviewer #2: Zika should be written with a capital Z as it is named after Zika forest.

Page 7 - Line 128 - remove apostrophe from OP's. It is better written as OPs (no possessive required).

Reviewer #3: My editorial suggestions:

Minor edits in the Introduction section:

-Lines 110-111: Knockdown resistance is defined, however, I think it would be useful to also indicate that knockdown is a characteristic effect of the pyrethroid insecticide class.

-Line 128: Remove the apostrophe for OPs (should be OPs, not OP's)

Minor edits in the Methods section:

-Line140: What was the material and color of the ovitraps used? Where they a specific brand or homemade?

-Line 146: At what approximate concentration (grams of diet per pan, etc.) and at what frequency were the larvae fed?

-Lines 165-175: What was the control treatment used for the LC50 testing? Was acetone alone used? Were the control mosquitoes untreated? Both? Please clarify.

-Line 184: The second H null here is written as 'Ho', please correct to H null similar to elsewhere in the paper. 

Minor edit to the Results section:

-Line 234: Please correct 'RR ... is shown' to 'RRs ... are shown'.

Minor edit to the Discussion section:

-Line 282: Please correct '[2014]' to '[21]'.

Minor edit to Figure 1:

-Please clarify the location of ALL the sites by placing a dot indicating each site. Some sites (e.g. Raymundo Enriquez, Paraiso, etc.) have the resistance ratios placed adjacent to the site, but the site location is not clear. 

Minor

**Summary and General Comments**

Reviewer #1: This manuscript by Solis-Santoyo, et al. describes a study of spatial variation in phenotypic and genotypic insecticide resistance in Aedes aegypti in the city of Tapachula, which has one of the highest incidences of dengue in Mexico. The authors report the discovery of significant spatial variation in insecticide resistance, including widespread, high-level pyrethroid resistance despite the discontinuation of pyrethroid use in the city 5 years prior to sample collection. Furthermore, one of the alleles that confers knockdown resistance to pyrethroids has nearly achieved fixation throughout Tapachula. The strengths of the study include the clarity of writing, detailed study design, and interesting findings that can inform the strategy of other mosquito control campaigns. The main weakness in the study is the absence of determining the relationship between distance of neighboring collection sites and insecticide resistance profile similarity, i.e. if neighboring mosquitos have more similar resistance phenotypes/genotypes than spatially distant non-neighbors. Related to this criticism is the use of a quadrant-based system to determine spatial effects, which has methodological limitations. Overall, however, this is a well-designed study with interesting findings, and I recommend minor revisions before it can be accepted for publication.

Reviewer #2: The paper is clearly written and the experimental approach seems rigorous. A little more attention to the published literature at the points mentioned above would improve the manuscript.

Reviewer #3: Overall, this was a well executed and analyzed study that will be useful not only to Mexico, but to other areas facing insecticide resistance issues with a limited complement of chemical control tools. The difference in the resistance to chlorpyrifos vs. malathion was intriguing and warrants a follow up study in a few years time.

PLOS authors have the option to publish the peer review history of their article (what does this mean?). If published, this will include your full peer review and any attached files.

Reviewer #1: Yes: Joshua R. Lacsina

Reviewer #2: No

Reviewer #3: Yes: Natasha M. Agramonte, PhD

Figure Files:

Data Requirements:

Reproducibility:

References

---

## [Decision Letter · Decision Letter 1]

19 Aug 2021

Dear Dr. Penilla-Navarro,

We are pleased to inform you that your manuscript 'Insecticide resistance in Aedes aegypti from Tapachula, Mexico: spatial variation and response to historical insecticide use' has been provisionally accepted for publication in PLOS Neglected Tropical Diseases.

Best regards,

Pattamaporn Kittayapong, Ph.D.

Associate Editor

Amy Morrison, Ph.D.

Deputy Editor

Reviewer's Responses to Questions

**Key Review Criteria Required for Acceptance?**

**Methods**

-Are the objectives of the study clearly articulated with a clear testable hypothesis stated?

-Is the study design appropriate to address the stated objectives?

-Is the population clearly described and appropriate for the hypothesis being tested?

-Is the sample size sufficient to ensure adequate power to address the hypothesis being tested?

-Were correct statistical analysis used to support conclusions?

-Are there concerns about ethical or regulatory requirements being met?

Reviewer #1: In the revised manuscript, the authors did an excellent job in clarifying the distinction between the spatial analysis performed using the 26 collection sites and the quadrant analysis to look for operational sources of variation in spraying activities. This makes much more sense, and strengthens the manuscript. I have no methodological concerns.

Reviewer #2: More information is needed about the blood-feeding of mosquitoes on rabbits. Were rabbits anaesthetised.? How long was each feeding event? Are there any ethical standards that you need to report to about this activity?

Reviewer #3: The objectives were clearly stated with an appropriately designed study to test the stated objectives. The population was appropriate and detailed well. The sample sizes were sufficient for the rigor of the statistical analyses used to support the conclusions. I discerned no ethical or regulatory concerns.

**Results**

-Does the analysis presented match the analysis plan?

-Are the results clearly and completely presented?

-Are the figures (Tables, Images) of sufficient quality for clarity?

Reviewer #1: Yes, by including the Moran's I correlogram in the revised manuscript, the analysis now clearly matches the analytic plan. Particularly with the addition of Figure 4, it is evident that the null hypothesis for correlating LC50s with space cannot be rejected, making this part of the analysis much clearer for the reader. I have no concerns about the results or analysis.

Reviewer #2: (No Response)

Reviewer #3: The analysis presented matched the methods stated. The results were clearly presented and the figures were sufficiently clear to elucidate the data described.

**Conclusions**

-Are the conclusions supported by the data presented?

-Are the limitations of analysis clearly described?

-Do the authors discuss how these data can be helpful to advance our understanding of the topic under study?

-Is public health relevance addressed?

Reviewer #1: Yes, the conclusions are supported by the data and the limitations of the analysis are clearly addressed. The impact and public health relevance of their discoveries are thoroughly discussed.

Reviewer #2: (No Response)

Reviewer #3: The conclusions appear to be supported by the data presented and the limitations noted are appropriate for this study. The authors discussed both the public health relevance and the usefulness of the study in advancing the field.

**Editorial and Data Presentation Modifications?**

Reviewer #1: No modifications recommended.

Reviewer #2: Line 61. Any use of insecticides may contribute to selection for insecticide resistance. Mismanagement of their use can have worse effects, but even with rational use, resistance can still develop.

Line 87: change to ‘As a control,’

Line 192: there is an unnecessary full stop after aegypti

Line 298: no apostrophe needed in LC50s.

Line 304: change to ‘were statistically significant’. This sentence is incomplete, but makes sense if it is joined to the next sentence with a comma …425 m), a caveat….

Line 309: change to ‘there is a very small number’

Line 363: change ‘resistant’ to ‘resistance’

Line 378: change ‘effectivity’ to ‘efficacy’

Line 384: change to ‘and, in fact, no mutations’

Line 395: does ‘More yet’ mean ‘Furthermore’?

Line 396: change to ‘based on their accessibility’ and ‘Assuming no spatial heterogeneity’

Line 398: change ‘de city’ to ‘the city’

Line 399: correct spelling of ‘aegypti’

Line 401: correct in-text reference

Line 404: change to ‘migration from sites that were untreated due to vehicle'

Line 408-9: This study needs a reference. This information has been stated earlier in the Discussion. Remove the repetition.

Reviewer #3: Accept

**Summary and General Comments**

Reviewer #1: This revised manuscript by Solis-Santoyo, et al. describes a study of spatial variation in phenotypic and genotypic insecticide resistance in Aedes aegypti in the city of Tapachula, which has one of the highest incidences of dengue in Mexico. The authors report the discovery of significant spatial variation in insecticide resistance, including widespread, high-level pyrethroid resistance despite the discontinuation of pyrethroid use in the city 5 years prior to sample collection. Furthermore, one of the alleles that confers knockdown resistance to pyrethroids has nearly achieved fixation throughout Tapachula. Importantly and contrary to the authors’ initial hypothesis, there was no correlation between LC50 and distance between collection sites despite largely uniform application of insecticide throughout the city. These results are both unexpected and intriguing as to the source of such high spatial variability. The revised manuscript is much improved due to (1) clarifying the reasons for and distinction between the spatial analysis and the quadrant analysis, and (2) the addition of the Moran’s I correlogram analysis. The authors have addressed all my critiques, and I recommend the manuscript be accepted for publication.

Reviewer #2: Most points identified in the first review have been changed appropriately.

The convention is a capital letter for Zika – Surely you are referring to Zika, the disease. Why are you being informal in a scientific paper? The journal editor can decide what is standard for this journal, but a brief perusal of articles shows the use of a capital letter by other authors in PloS NTD.

Reviewer #3: The clarity and rigor of the manuscript is much improved. Nominal minor edits include:

-Line 331, 353-354: Type 1 and/or Type 2 typically precede the pyrethroid (e.g. Type 1 pyrethroid)

-please italicize 'kdr' throughout the manuscript

-Line 401: this appears to be a reference formatting typo and (37 GROSSMAN) should be changed to [37]

PLOS authors have the option to publish the peer review history of their article (what does this mean?). If published, this will include your full peer review and any attached files.

Reviewer #1: **Yes: **Joshua R. Lacsina

Reviewer #2: No

Reviewer #3: **Yes: **Natasha Marie Agramonte

---

## [Editor Report · Acceptance letter]

6 Sep 2021

Dear Dr. Penilla-Navarro,

We are delighted to inform you that your manuscript, "Insecticide resistance in Aedes aegypti from Tapachula, Mexico: spatial variation and response to historical insecticide use," has been formally accepted for publication in PLOS Neglected Tropical Diseases.

Best regards,

Shaden Kamhawi

co-Editor-in-Chief

Paul Brindley

co-Editor-in-Chief
